# Surrogate Modeling for Computationally Expensive Simulations of Supernovae in High-Resolution Galaxy Simulations

**Keiya Hirashima**[1]*,  **Kana Moriwaki**[1],  **Michiko S. Fujii**[1],  **Yutaka Hirai**[2,3],
**Takayuki R. Saitoh**[4],  **Junichiro Makino**[4],  **Shirley Ho**[5,6,7,8]
[1]The University of Tokyo,   [2]University of Notre Dame,   [3]Tohoku University,   [4]Kobe University,
[5] New York University,   [6] Flatiron Institute,   [7] Princeton University,   [8]Carnegie Mellon University

## Abstract

Some stars are known to explode at the end of their lives, called supernovae (SNe). The substantial amount of matter and energy that SNe release provides significant feedback to star formation and gas dynamics in a galaxy. SNe release a substantial amount of matter and energy to the interstellar medium, resulting in significant feedback to star formation and gas dynamics in a galaxy. While such feedback has a crucial role in galaxy formation and evolution, in simulations of galaxy formation, it has only been implemented using simple *sub-grid models* instead of numerically solving the evolution of gas elements around SNe in detail due to a lack of resolution. We develop a method combining machine learning and Gibbs sampling to predict how a supernova (SN) affects the surrounding gas. The fidelity of our model in the thermal energy and momentum distribution outperforms the low-resolution SN simulations. Our method can replace the SN sub-grid models and help properly simulate un-resolved SN feedback in galaxy formation simulations. We find that employing our new approach reduces the necessary computational cost to $\sim 1$ percent compared to directly resolving SN feedback.

## 1   Introduction

Supernova (SN) explosions are highly energetic events that release immense energy that heats and expels the ambient interstellar medium (ISM). This results in galactic outflows and turbulence, influencing star formation rates and galaxy scale heights [1]. SNe significantly impact galaxy evolution, incorporating processes like gravity, hydrodynamics, radiation, star formation, and chemical changes. Given the complicated interplay among these processes, numerical methods have been commonly employed to study galaxy formation and evolution.

In simulations of the Milky Way Galaxy (MW) using Lagrangian methods, stars, and dark matter are modeled through $N$-body simulations, and the gas dynamics is modeled using Lagrangian (particle-based) methods, such as smoothed particle hydrodynamics (SPH). Compared to Eulerian (mesh-based) methods, SPH offers advantages for galaxy simulations. For instance, SPH's Galilean invariance handles non-symmetric systems, like a galaxy's ISM. Moreover, SPH avoids numerical diffusion issues, which arise when the advection isn't aligned with the mesh, ensuring accurate mass conservation even when managing the complex multi-phase ISM. In the $N$-body/SPH, individual particles represent a clump of dark matter, gas, or a group of stars, and it is desirable to use particles with as small mass as possible. In the state-of-the-art simulations, the mass resolution has reached $\sim 10^3$ $M_\odot$[1] [2, 3]. Further higher mass resolution is necessary to resolve the contribution of SNe

---

*hirashima.keiya@astron.s.u-tokyo.ac.jp

[1]1 $M_\odot$ (solar mass) is a unit of mass equal to that of the Sun.

Neural Information Processing Systems (NeurIPS 2023) AI for Science Workshop

properly [4, 5]. To push forward the mass resolution to individual stars, i.e., $\sim 1$ M$_\odot$, we are developing an $N$-body/SPH code for galaxy formation and evolution, ASURA-FDPS [6, 7]. Since such simulations consider multi-scale physics and are performed with supercomputers, a tiny fraction of short timescale regions become bottlenecks for large-scale parallel computations due to Amdal's law. In hydrodynamics simulations, timescales become shorter in high-temperature, density, and pressure environments (e.g., supernova explosions i.e., SNe). We develop a machine learning model that learns from SN simulations in turbulent gas clouds with $1$ M$_\odot$ resolution, to reduce the simulation cost by replacing direct numerical simulations of short time-step regions induced by SNe with the prediction from machine learning.

Here, we adopt a *surrogate model* that learns the relationship between the initial condition $x$ of the simulation and the result $y$ calculated by direct numerical simulations (DNS) for a certain time in the field. The surrogate model can predict results in fewer steps than DNS. Recent studies have developed convolutional neural network (CNNs)-based models to model turbulence [e.g., 8, 9, 10] for Eulerian simulations at field level. Compared to Eulerian simulations, which use voxel-wise data, applying CNNs to Lagrangian simulations is not straightforward. Nevertheless, there has been recent research on learning time evolution with CNNs in Lagrangian simulations by interpolating the particles onto voxels [e.g., 11, 12, 13]. Motivated by these studies, we develop a CNNs-based model that learns the relation between two distributions in voxels interpolated from SNe simulations in SPH.

## 2   Background

This section describes the motivations for accelerating this supernova feedback and briefly describes how one generates these training sets by direct supernova simulations. The setup for the simulations is similar to [13].

**Bottlenecks in galaxy simulations**   Scaling Milky Way (MW)-sized galaxy simulations to represent each star at a $1$ M$_\odot$ resolution introduces computational challenges. Such simulations demand processing over $10^{10}$ gas and star particles, two orders larger than current state-of-the-art simulations can handle. A core challenge is ensuring short timesteps for precision, especially in high-density and temperature regions. Recent galaxy simulations have adopted a hierarchical timestep method to reduce redundant computations. To illustrate, particles influenced by SNe, which are just a minor fraction of the galaxy, use much shorter timesteps than the rest. The majority can be managed with a longer, uniform timestep. However, even with the hierarchical timestep method in place, galaxy simulations encounter a significant challenge in communication overhead. For each minor timestep iteration, particles requiring it must be transferred among computation nodes. This frequent communication results in many idle CPU cores during these transfer periods. As the number of processors or steps amplifies, this communication overhead grows more pronounced, leading to decreased scalability. Therefore the scaling cannot be improved even though supercomputers are advanced and have a larger number of on-board CPU cores. This emphasizes the importance of surrogate models to quickly reconstruct the results requiring short steps (e.g., SNe) to improve these large galaxy simulations' computational efficiency and scalability.

**Setup for Simulation of training data**   We simulate SNe using ASURA-FDPS [6, 7], considering self-gravity, hydrodynamics, and radiative cooling. The code implements the hydrodynamics with the smoothed particle hydrodynamics (SPH) method. The SPH has widely been used in astrophysical and fluid dynamics simulations to solve compressible fluid and multi-scale physics. This method discretizes a fluid into finite mass particles. It evaluates hydrodynamic interactions among neighboring particles to update the motion and state. At each timestep, physical quantities $f_i$ of the particle $j$ are evaluated as follows:

$$f_i = \sum_j m_j \frac{f_j}{\rho_j} W(r_{ij}, h), \tag{1}$$

where $m_j$, $\rho_j$, $r_{ij}$, $h$, and $W(r_{ij}, h)$ are the mass, density, distance between the particle $i$ and $j$, and a smooth function. The SPH method may encounter difficulties in resolving contact discontinuities caused by shock waves (such as SN shells) with low mass resolution. Nevertheless, the code has been tested and verified to resolve the shock wave accurately. It can capture the formation of SN shells caused by thermal energy when the mass resolution is finer than $1$ M$_\odot$ [13].

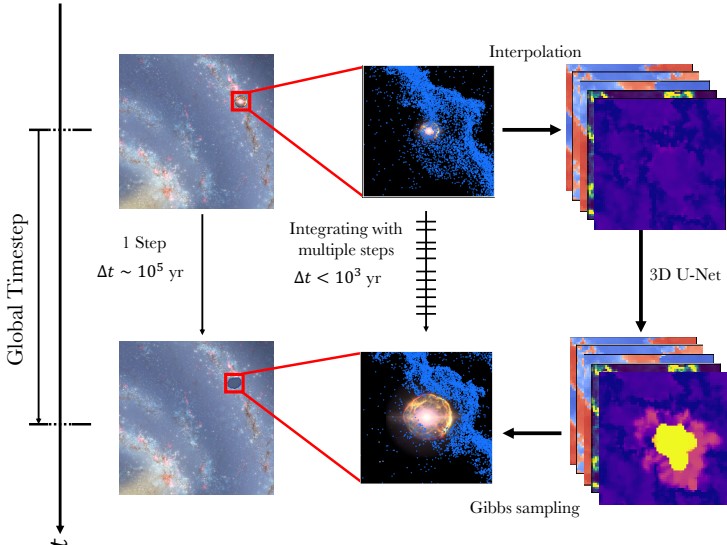

Figure 1: Schematic overview of our method. *Left*: the galaxy's spiral arm in snapshots at two global timesteps. *Center*: the region with a short timescale due to the SN. *Right*: Our method and prediction process in voxels. Image credit:[*left*: NASA/JPL-Caltech/ESO/R. Hurt, *middle*: ESA/Hubble (L. Calçada).]

We initially evolved homogeneous gas spheres with a turbulent velocity field that follows $\propto v^{-4}$ for one initial free fall time to make gas clouds similar to star-forming regions in the MW. Then our simulations inject an explosion energy of $10^{51}$ erg into 100 SPH particles (100 $M_\odot$) at the center of turbulent gas clouds. This thermal energy is distributed among the particles with weights by SPH kernels. In our simulations, SNe explode at the center of mass in these turbulent gas clouds.

## 3 Method

Fig. 1 shows the schematic overview of our method. The direct numerical simulations (DNS) for SN regions in the center column need shorter timesteps than $\sim 10^3$ years. This expensive process causes frequent calculation and communication steps, which can be a bottleneck in parallel computing. This section describes our three separate methods, SPH interpolation, U-Net [14], and Gibbs sampling, for bypassing this process and emulating DNS results faster.

**Preprocessing: SPH interpolation method**  To be able to handle particle data in CNNs-based models, physical quantities of SPH particles are interpolated in voxels with 3D Cartesian coordinates. Specifically, the density, temperature, and 3D velocities are represented as five 3D scalar fields of size $64^3$ voxels. One side of the box is 60 parsec. The SPH particles are interpolated using the SPH kernel convolution using equation (1). The number of neighbor particles is set to be 64. Normalizing interpolation can improve the precision of quantities interpolated into voxels by eliminating the effects related to particle distribution [15]. Quantities are normalized by making the total of interpolation weights $w(i)$ unity. The weight $w(i)$ of particle $i$ for equation (1) is defined as the following:

$$
\begin{aligned}
w(i) &:= \sum_{j=1}^{N} \frac{m_j}{\rho_j} W(|\boldsymbol{r}_i - \boldsymbol{r}_j|, h_i) \\
&= \sum_{j=1}^{N} \frac{m_j}{\rho_j} W_{i,j}.
\end{aligned}
\tag{2}
$$

Dividing equation (1) with equation (2), normalized quantities $\tilde{f}(i)$ (i.e., temperature and velocities) are written as follows:

$$
\tilde{f}(i) := f(i)/w(i).
\tag{3}
$$

The density distribution must retain particle distribution information and is therefore not subject to the normalization with equation (3).

Compressible hydrodynamics simulations have a wide range of physical quantities. To have the model effectively learn the dataset of the simulations, we normalized density and temperature:

$$\rho^* := \log_{10} \rho, \tag{4}$$

$$T^* := \log_{10} \tilde{T}. \tag{5}$$

Velocities have a bimodal distribution. To improve the accuracy, the 3D velocities were distributed into six colors.

$$v^*_{.,p} := \begin{cases} \log_{10} \tilde{v}. & \text{if } \tilde{v}. > 0 \\ 0 & \text{if } \tilde{v}. <= 0 \end{cases} \tag{6}$$

$$v^*_{.,n} := \begin{cases} 0 & \text{if } \tilde{v}. >= 0 \\ \log_{10}(-\tilde{v}.) & \text{if } \tilde{v}. < 0 \end{cases} \tag{7}$$

where $v. \in \{v_x, v_y, v_z\}$. Given by these conversion, these eight features, $\{\rho^*, T^*, v^*_{x,p}, v^*_{x,n}, v^*_{y,p}, v^*_{y,n}, v^*_{z,p}, v^*_{z,n}\}$, are used in training.

**3D U-net** We use a model that extends U-net [14]. The internal dimension is extended from 2D to 3D. The channel is set to be five. The mean squared error (MSE) is used for the loss function with an equal weight for each channel. The loss is minimized with ADAM optimizer [16] with a learning rate of $10^{-5}$. The model $\mathcal{M}$ is trained to learn the relation between input $X$ and output $y$, where $X$ and $y$ represent the distribution of physical quantities before and after the SN explosion in simulations, respectively, over a period of one hundred thousand years. Suppose the trainable parameter $\theta$, the predicted distribution $\hat{y}$ is written as the following:

$$\hat{y} = \mathcal{M}(X \mid \theta). \tag{8}$$

**Postprocessing: reconstruction to the distribution of SPH particles** Particles are sampled using Gibbs sampling, which is a Markov chain Monte Carlo method. Gibbs sampling can generate an approximate sequence of samples by using conditional probabilities of one variable. Using the predicted distribution of density $\hat{\rho}$, the probability is written as

$$p(x, y, z) := \frac{\hat{\rho}(x, y, z)}{\sum_x \sum_y \sum_z \hat{\rho}(x, y, z)}, \tag{9}$$

as the joint probability where $x, y, z$ are the positions of voxels. The positions of the particles are determined by adding perturbations to avoid overlap, preventing physically unrealistic situations. The perturbation values are drawn from a uniform distribution within the range $[0, \Delta w]$, where $\Delta w$ corresponds to the width of one voxel. This study defines $\Delta w$ as approximately 1 parsec, calculated as $\Delta w = 60$ parsec $/64 \sim 1$ parsec. The burn-in period is set to be 1000. Given a joint distribution $p(x, y, z)$, the Gibbs sampling generates samples as follows:

1. Choose an initial state $(x^{(0)}, y^{(0)}, z^{(0)})$.
2. For each iteration $t = 1, 2, \ldots, N$:
   (a) Sample $x^{(t)}$ from $p\left(x \mid y^{(t-1)}, z^{(t-1)}\right)$.
   (b) Sample $y^{(t)}$ from $p\left(y \mid x^{(t)}, z^{(t-1)}\right)$.
   (c) Sample $z^{(t)}$ from $p\left(z \mid x^{(t)}, y^{(t)}\right)$.

By setting $N$ as the number of particles in the box, mass conservation is ensured before and after the prediction.

In the context of SN explosions in galaxy evolution, it is crucial to accurately reconstruct both thermal energy and radial outward momentum [e.g., 17]. However, given that these carriers constitute only a fraction of the particles in the voxels, increasing the number of sampling iterations becomes essential to attain high accuracy. Optimizing the number of sampling iterations is necessary to improve computational efficiency, which is achieved through implementing a temperature-based condition. Fig 2 shows the relation between the mean density at the center at $t = 0$ from the explosion and the

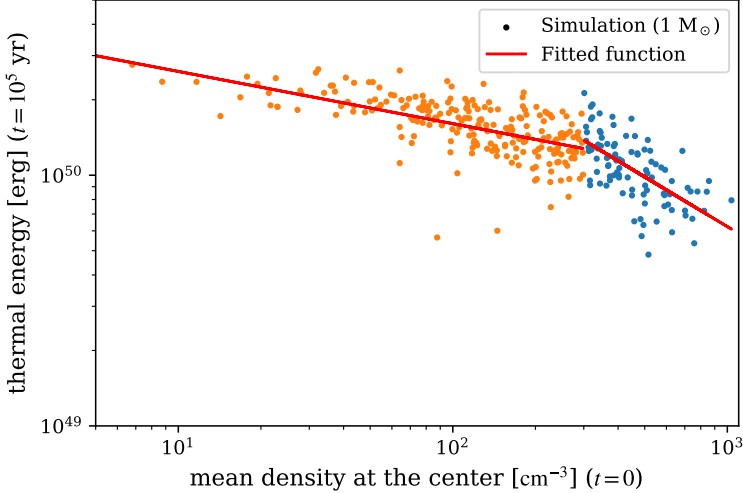

Figure 2: The relation between the mean density at the center before and the thermal energy $10^5$ years after the SN explosions. The piece-wise linear function was fitted using the training dataset of the 3D U-Net. The horizontal axis breakpoint is at $300 \ \mathrm{cm}^{-3}$. Orange (blue) points indicate samples with low (high) initial density at the center.

thermal energy at $t = 10^5$ years. The piece-wise linear function was fitted using the training dataset. The breakpoint at $300 \ \mathrm{cm}^{-3}$ on the horizontal axis is determined by maximizing the determination coefficients of the fitted functions. After Gibbs sampling, the thermal energy of the reconstructions is compared to the estimated thermal energy to determine if sampling should continue. Sampling is repeated until the difference is within a threshold. In this test, the threshold is 5 percent, and the iteration occurs only a few times.

# 4 Experiments and results

## 4.1 Morphology

Fig 3 shows the initial condition, a DNS's result with SPH and 1 $M_\odot$ resolution, the prediction by our model, and the DNS's result with SPH and 10 $M_\odot$ resolution. The deep learning model takes the 3D distribution (voxels) of density, temperature, and 3D velocities and predicts those that elapsed one hundred thousand years, which is equivalent to the global timestep in galaxy simulations. The model reconstructs the asymmetric shell and hot region. It also reconstructs the global features of velocities: for instance, on the x-axis, $v_x$ becomes positive on the positive area from the center, and it becomes negative on the negative side. It is still challenging to resolve the complex velocity field at the center.

Fig 4 shows the (a) initial condition, (b) simulation elapsed $10^5$ years, (c) reconstruction generated by sampling particles from the predicted 3D physical quantities, and (d) simulation elapsed $10^5$ years with a low resolution. The reconstruction is globally similar to the simulation but more blurred due to the limitation of spatial resolution. Comparing (b) and (d), the low-resolution simulation shows that the thin, dense shell of the SN is not resolved well.

## 4.2 Fidelity

In the evolution of galaxies, SNe heat and disperse the surrounding gas, thereby introducing outward momentum to the ambient gas and suppressing star formation. Therefore, we evaluate the fidelity of our approach in thermal energy and radial outward momentum using the simulations with a lower mass resolution (10 $M_\odot$). The low-resolution simulations (10 $M_\odot$) are computed using the same initial turbulence field as in the corresponding high-resolution case (1 $M_\odot$). Fig. 5 shows the discrepancy in the thermal energy between the simulations with high-resolution and low-resolution (*left*) and the simulations with high-resolution and our approach (*right*). Ideally, the thermal energy of the low-resolution simulations and our approach must converge to that of the high-resolution

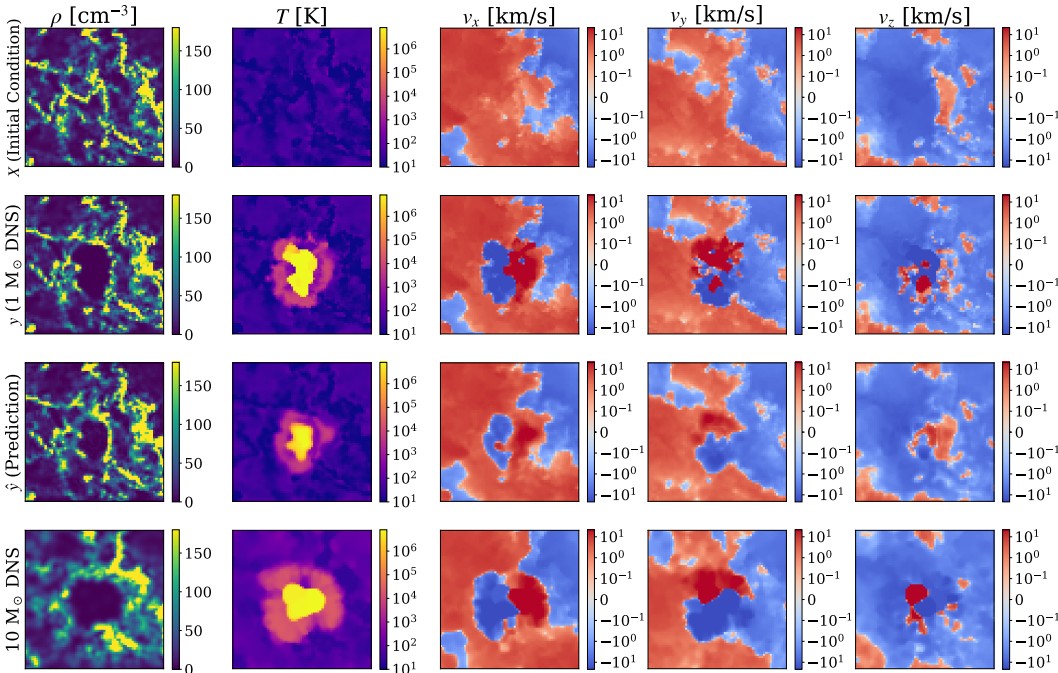

Figure 3: The prediction of the 3D U-Net. We show the initial condition, the result of high-resolution (1 M$_\odot$) DNS, the prediction by the 3D U-Net, and the result of low-resolution (10 M$_\odot$) DNS from top to bottom row. Cross sections of density $\rho$, temperature $T$, and 3D velocity ($v_x$, $v_y$, $v_z$) are lined up from left to right.

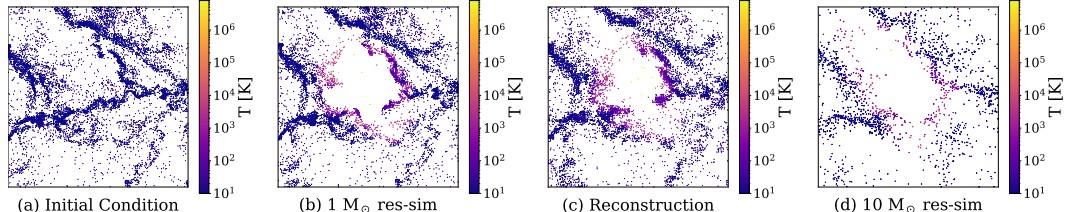

(a) Initial Condition   (b) 1 M$_\odot$ res-sim   (c) Reconstruction   (d) 10 M$_\odot$ res-sim

Figure 4: (a) The initial condition just before a SN explodes. (b) The result of high-resolution (1 M$_\odot$) DNS $10^5$ years after the explosion. (c) The reconstruction $10^5$ years after the explosion by our approach. (d) The result of low-resolution (10 M$_\odot$) DNS $10^5$ years after the explosion. The color bar represents the temperature $T$ of each particle.

simulation. Here, the discrepancy is evaluated with the determination coefficient $R^2$, root mean square error (RMSE), and mean absolute percentage error (MAPE). Comparing these indices, we found our approach can reconstruct the thermal energy of the high-resolution simulations. Fig. 6 shows the discrepancy in the radial outward momentum. The settings are the same as the evaluation of the thermal energy. Our approach is slightly better at reconstructing the radial outward momentum than the low-resolution simulations.

### 4.3 Discussion

**Speed-up**   We compare the required time between DNS with the high-resolution (1 M$_\odot$) and our approach, which consists of interpolation, the 3D U-net's prediction, and Gibbs sampling, using one calculation node on the supercomputer Fugaku. While the DNS needs several hours to evolve the SN explosion for $10^5$ years, the interpolation and Gibbs sampling take a few seconds. Although the prediction would be a bottleneck, optimized by SOFTNEURO®[2] [18], it can be done within 0.2

---

[2]SOFTNEURO® is developed by Morpho, Inc.

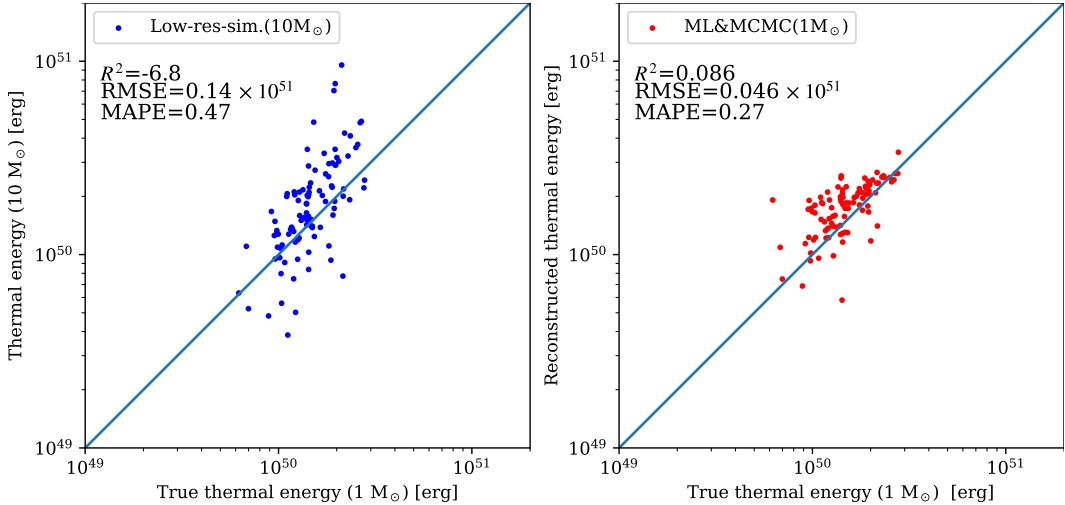

Figure 5: Fidelity evaluation in thermal energy. Using the high-resolution simulation (1 $M_\odot$ resolution; $x$-axis) results as a baseline, we compared our method ($y$-axis, *right*) with the corresponding low-resolution simulation (10 $M_\odot$ resolution; $y$-axis, *left*). We evaluate 100 test data by the determination coefficient $R^2$, root mean squared error (RMSE), and mean absolute percentage error (MAPE).

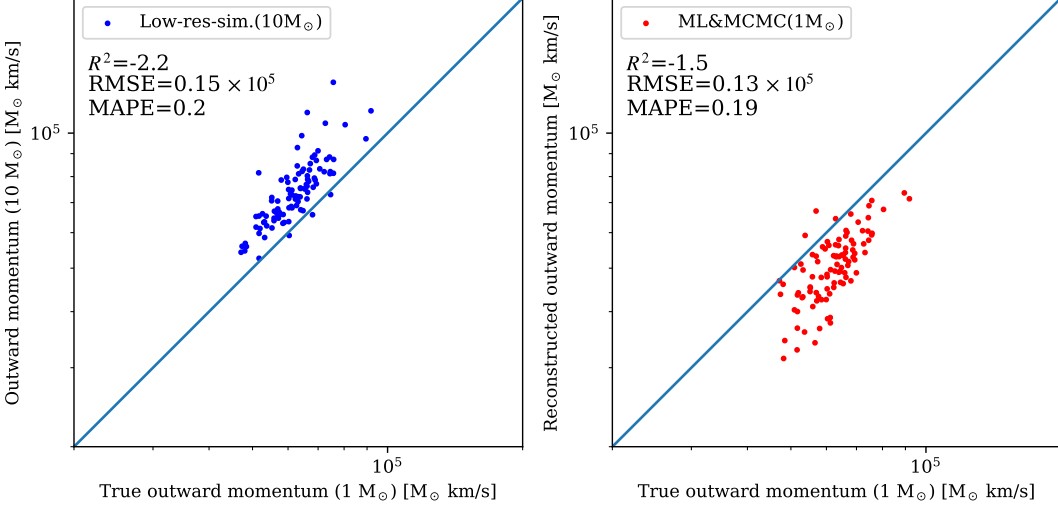

Figure 6: Fidelity evaluation in radial outward momentum. Other settings are the same as Fig 5.

seconds with one node. While the DNS takes a few node hours, our approach does node a few seconds overall. The calculation costs are reduced to less than 1 percent.

**Possible improvements in resolution**   Enhancing resolution during reconstruction is crucial to achieve more precise prediction results. First of all, coarse simulations might not accurately capture physical properties. For example, the low-resolution ($10$ $M_\odot$) simulation fails to resolve the thin and dense shells that are visible in the high-resolution simulation ($1$ $M_\odot$), as seen in Fig. 3 and 4. Additionally, our model's prediction may be negatively affected if there are only a few voxels with high energy and temperature due to the low resolution, and the model could fail to predict them. This can cause a significant error between the ground truth and prediction. In fact, as we discussed in Section 4.2, predicting momentum is more challenging than thermal energy since a voxel with large momentum is less than a voxel with large thermal energy. Furthermore, coarse resolution negatively impacts the performance of Gibbs sampling. In cases where only a few particles carry high momentum and temperature, the accuracy easily worsens if the sampling misses those particles. To overcome these challenges, future research will use simulations with enhanced resolution.

When enhancing voxel resolution, it is important to adjust the number of particles in the original particle data to prevent the creation of small meshes that require higher resolution than the initial particle data. For example, if the number of voxels is doubled on one axis, the number of particles should also increase by approximately 10 (approximately $2^3$). In our study, a resolution of $128^3$ meshes with $0.1$ $M_\odot$ for mass resolution will be appropriate. It is crucial to consider these factors when improving a voxel resolution and to plan for future developments.

## 5   Conclusions and future directions

We have developed the first surrogate model for SN feedback toward high-resolution galaxy formation simulations. We implement a pipeline of particle interpolation and sampling from voxels to utilize CNN-based models that have been successfully used to learn and predict multiple astronomical simulations. The multiple test datasets are used to evaluate the conservation of thermal energy and outward momentum. We find that our approach can produce results that are more consistent than low-resolution simulations ($10$ $M_\odot$), which is sufficiently high compared to the mass resolution of current galaxy formation simulations. However, improvements can still be made with respect to momentum conservation. In future studies, we can improve the predictive performance of machine learning by using even higher-resolution supervised data. In addition, we will study the method to project the prediction of the model learning high-resolution calculations onto low-resolution simulations. Such a method may be able to improve low-resolution simulations of galaxy scale.

## 6   Acknowledgement

Numerical computations were carried out on Cray XC50 CPU-cluster at the Center for Computational Astrophysics (CfCA) of the National Astronomical Observatory of Japan and Flatiron Institute's Rusty computing cluster. The model was trained using Wisteria/BDEC-01 at the Information Technology Center at the University of Tokyo. This work was also supported by JSPS KAKENHI Grant Numbers 22H01259, 22KJ0157, 20K14532, 21H04499, 21K03614, 23K03446, and, 22J23077, MEXT as "Program for Promoting Researches on the Supercomputer Fugaku" (Structure and Evolution of the Universe Unraveled by Fusion of Simulation and AI; Grant Number JPMXP1020230406), and Initiative on Promotion of Supercomputing for Young or Women Researchers, Supercomputing Division, Information Technology Center, The University of Tokyo. One of the authors is financially supported by the JSPS Research Fellowship for Young Scientists, JSPS Overseas Challenge Program for Young Researchers, JEES · Mistubishi Corporation science technology student scholarship in 2022, and the IIW program of The University of Tokyo.

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

## Appendix

## A  Example results of the predcition

We show other prediction examples and results of corresponding low-resolution simulations.

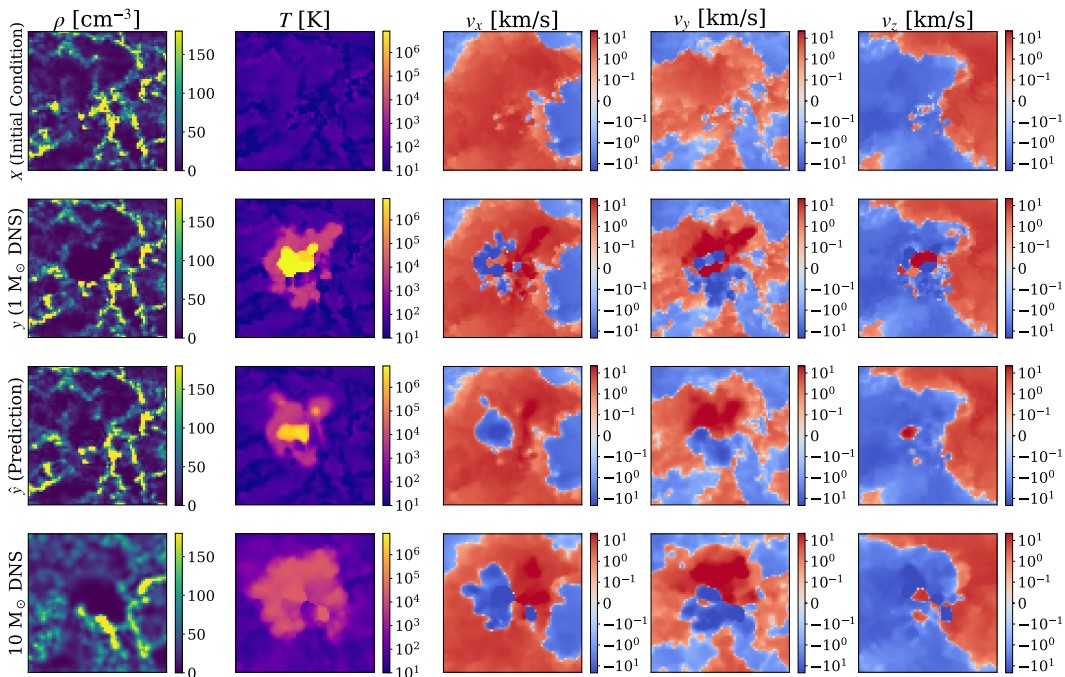

Figure 7: Physical quantities and colorbar mean the same as Fig. 3.

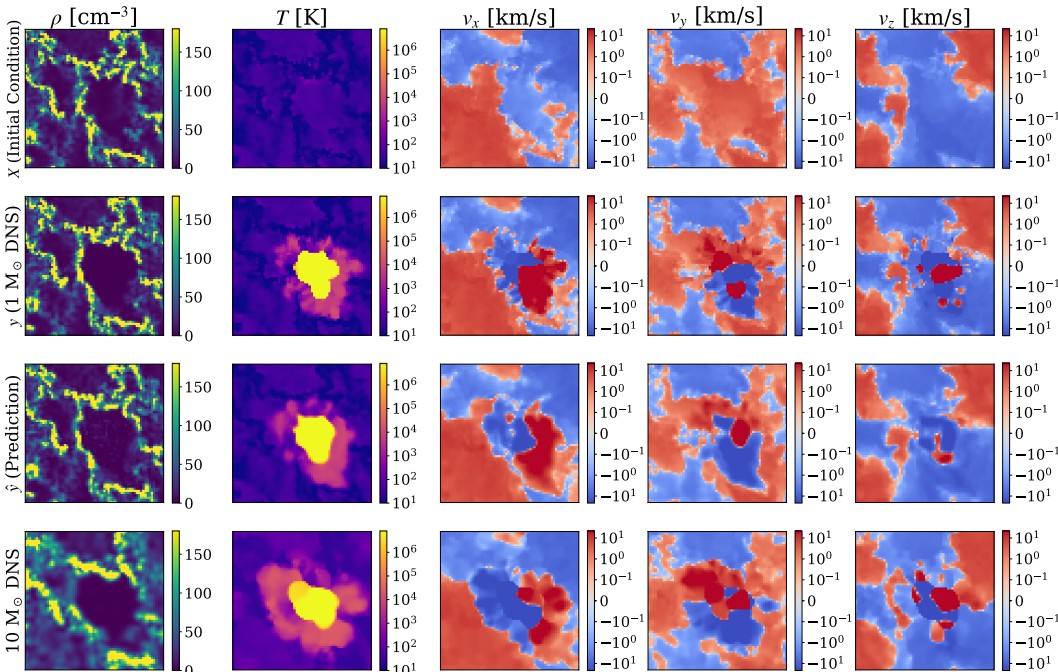

Figure 8: Physical quantities and colorbar mean the same as Fig. 3.

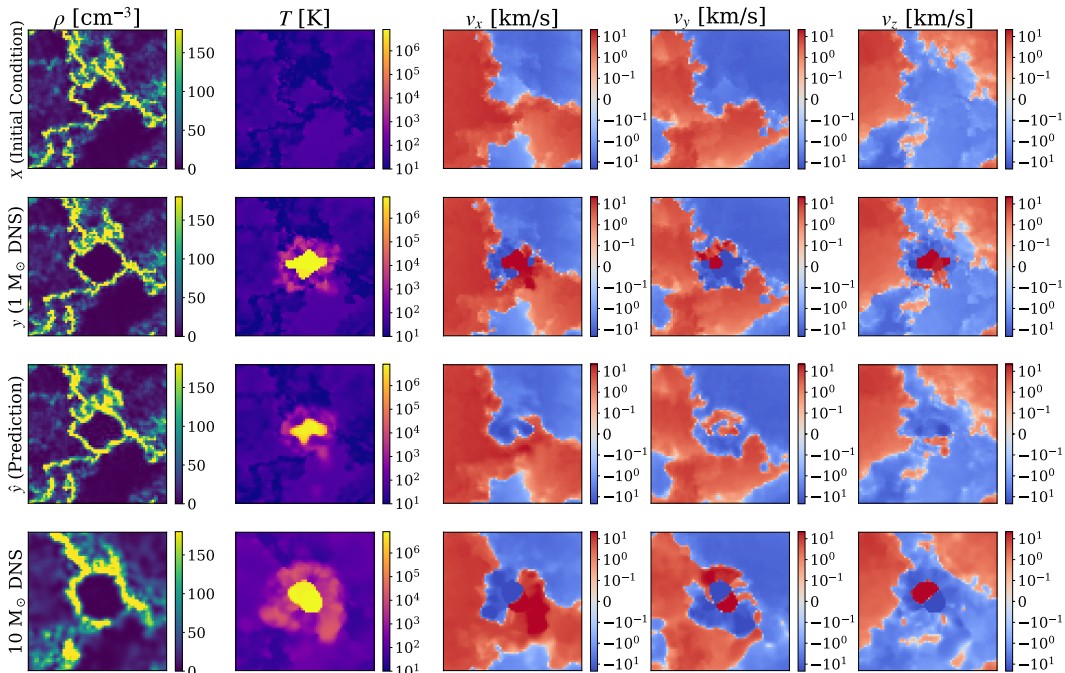

Figure 9: Physical quantities and colorbar mean the same as Fig. 3.

