# OpenReview forum: "Surrogate Modeling for Computationally Expensive Simulations of Supernovae in High-Resolution Galaxy Simulations"
_NeurIPS.cc/2023/Workshop/AI4Science — NeurIPS2023-AI4Science Poster_

### Official Review · Reviewer_X5Eq · 2023-10-22
**Good Application of AI4Science, but work needs polishing**

**Rating:** 6
**Confidence:** 2

**Review:**

Strengths

Because of the computational complexity of these simulations, I think this is a great use of machine learning in the sciences.

Line 92: The three sections about the different methods is really helpful.

The Wall Clock improvement described in section 4.3 is great.

Weaknesses

Line 78: Are all the variables addressed/described? What is h?
Line 83: What is v^-4?
Line 109: What are the six colors? For all this featurization, can you provide some intuition about why the featurization is necessary?

Figure 4: Are the three plots to the right supposed to have anything in them? They are blank for me.

Figures 5 & 6: How is the R^2 not bounded by 0-1? Here they are even negative. Perhaps this is a different metric than the coefficient of determination? It would also be helpful to know the approximate ranges of momentums and thermal energies to evaluate if the RMSEs and MAPEs are good or bad.

Some formatting issues:
The solar mass subscript should be included when it is first introduced in the introduction, not the background.
Is the “Bottlenecks in galaxy simulations” actually Amdals’ law? Should that be addressed in the background too?

---

### Official Review · Reviewer_UMJG · 2023-10-25
**Good Paper but needs more improvement**

**Rating:** 5
**Confidence:** 4

**Review:**

Summary:

The paper tackles the problem of studying the correlation between supernovae and star formation dynamics. Motivated by the heavy and time-consuming simulation methods, the authors propose employing machine learning to help advance and accelerate research on galaxy formation. Particularly, through ML-based methods, the simulation cycles can be replaced by surrogate models that can predict the characteristics of the star and its surroundings from data characterizing the supernovae. The approach has been implemented in three steps:
Preprocessing: selecting the physical quantities from the supernovae and applying normalization and data transformation on the data.
Model formulation: by employing a 3D U-Net.
Post-processing: by sampling particles using the Gibbs method.

Generally, the paper is well-written and discusses a relevant problem to science, astronomy, and astrophysics. Although the ultimate goal is fixed from the beginning (i.e., minimize simulation time), the approach is not well-motivated throughout the paper. Specifically, some choices on the used model (i.e., U-Net), Gibbs sampling, and feature selection have not been discussed in detail.

Furthermore, the paper lacks a comparison with related works. Surrogate modeling has been widely used in many fields. To the best of my knowledge, this is not the first paper that proposes surrogate models for star and galaxy formation prediction. A “Related works” section is missing to help the reader understand the particularity and contribution of this work.

Finally, the evaluation section needs to be more organized. Details on the used dataset, training-validation splitting, U-Net hyperparameters, and sensitivity analysis are missing. The obtained results are not discussed in detail, which leaves many questions unanswered. For instance, how can the model be generalized on unseen physical quantities and particles? How much time does the U-Net training take?

Overall, the paper tackles a relevant and critical scientific problem but needs more time to get it in good shape. I recommend structuring the “approach” and “evaluation” sections by explaining the authors' choices and conducting a more comprehensive evaluation, sensitivity analysis, and ablation study.


Strengths:
- The paper tackles a critical issue in star formation simulation.
- Adopting a surrogate model approach would benefit and accelerate the research in galaxy formation by replacing simulation cycles with ML-based surrogate models.


Weaknesses:
- The obtained results are not comprehensively discussed or investigated in detail. For instance, the generalization of the model wasn’t discussed in the paper.
- The paper lacks quantitative and qualitative comparison to state-of-the-art methods.
- The employment of the U-Net and Gibbs sampling are not well motivated.
- Figure 4 is not complete. There are no results to analyze.


Other minor comments:
- The abstract contains a repetitive expression “feedback to star formation and gas dynamics in a galaxy”. Consider rephrasing it in the second usage.
- Typo: “..by using conditional probabilities of one valuable.” → “..by using conditional probabilities of one variable”.
- Figure captions need to be improved.

---

### Meta-Review · Area_Chair_pW2K · 2023-10-27

**Recommendation:** Accept (Poster)
**Confidence:** 3

**Metareview:**

The authors describe a method to explore the problem of relating supernovae with star formation using machine learning to speed up the calculations. It addresses an interesting topic and presents some new approaches which will be of interest. There are some issues with the which it is recommended for the authors to address and correct before final submission, including a Related Works section and comparison to existing methods, correction of some metrics and figures, and more details on experimental choices such as dataset and model decisions, to allow reproducibility.